# SynEVO: A neuro-inspired spatiotemporal evolutional framework for cross-domain adaptation

Jiayue Liu [1]   Zhongchao Yi [1]   Zhengyang Zhou [1 2 3]   Qihe Huang [1]   Kuo Yang [1]   Xu Wang [1 2]   Yang Wang [1 2]

## Abstract

Discovering regularities from spatiotemporal systems can benefit various scientific and social planning. Current spatiotemporal learners usually train an independent model from a specific source data that leads to limited transferability among sources, where even correlated tasks requires new design and training. The key towards increasing cross-domain knowledge is to enable collective intelligence and model evolution. In this paper, inspired by neuroscience theories, we theoretically derive the increased information boundary via learning cross-domain collective intelligence and propose a Synaptic EVOlutional spatiotemporal network, SynEVO, where SynEVO breaks the model independence and enables cross-domain knowledge to be shared and aggregated. Specifically, we first re-order the sample groups to imitate the human curriculum learning, and devise two complementary learners, elastic common container and task-independent extractor to allow model growth and task-wise commonality and personality disentanglement. Then an adaptive dynamic coupler with a new difference metric determines whether the new sample group should be incorporated into common container to achieve model evolution under various domains. Experiments show that SynEVO improves the generalization capacity by at most 42% under cross-domain scenarios and SynEVO provides a paradigm of NeuroAI for knowledge transfer and adaptation. Code available at https://github.com/Rodger-Lau/SynEVO.

## 1. Introduction

Spatiotemporal learning aims to predict future urban evolution, which facilitates urban management and socioeconomic planning. Recently, diverse spatiotemporal forecasting solutions have well resolved data sparsity (Zhou et al., 2020), temporal shifts (Zhou et al., 2023), as well as unseen area inferences (Feng et al., 2024). As usual practices, almost all spatiotemporal learners train independent models from specific sources where both models and data are isolated. With the growing of available urban sensors and the diversity of data sources under the rapid urban expansion, the task-specific data-driven learning is inevitably faced with the increased costs of repetitive model designs and computational resources. To this end, an evolvable data-adaptive learner that accommodates cross-domain transferability and adaptivity is highly required to facilitate sustainable urban computing. There have been a number of efforts that try to improve generalization of spatiotemporal learning, where it can be classified as two folds, i.e., countering shifts on the same source domain and across different source domains. On the same source, the initial step is the continuous spatiotemporal learning implemented with experience reply (Chen et al., 2021), and spatiotemporal out-of-distribution issue (OOD) is raised by capturing causal invariance for confronting temporal shifts (Zhou et al., 2023). After that, a series of models take environments as indicators to guide generalization (Xia et al., 2024; Wang et al., 2024a; Yuan et al., 2023). However, even for different kinds of sources in a same system, these independent models fail to share common information for task transfer across domains. For generalization across sources, a prompt-empowered universal model for adapting various data sources (Yuan et al., 2024), and a task-level continuous spatiotemporal learner, which actively capture the stable commonality and fine-tune with individual personality for new tasks (Yi et al., 2024) are proposed. Even so, these models still suffer three critical issues for cross-domain transfer and data adaptive model evolution. 1) **No theoretical guarantee** for collective intelligence facilitating cross-domain transfer. 2) Not all tasks share common patterns, thus uniformly involving all tasks inevitably **introduces noise**. 3) These models are **not elastic** to actively evolve when data distribution changes.

[1]University of Science and Technology of China (USTC), Hefei, China [2]Suzhou Institute for Advanced Research, USTC, Suzhou, China [3]State Key Laboratory of Resources and Environmental Information System, Beijing, China. Correspondence to: Zhengyang Zhou <zzy0929@ustc.edu.cn>, Yang Wang <angyan@ustc.edu.cn>.

*Proceedings of the 42nd International Conference on Machine Learning*, Vancouver, Canada. PMLR 267, 2025. Copyright 2025 by the author(s).

Fortunately, with the progress of Neuro-Artificial Intelligence (NeuroAI), neural networks are designed to imitate knowledge acquiring for model generalization and cooperation (Wang et al., 2022; 2024b), such as neuro-inspired continuous learning (Wang et al., 2023b)and complementary-based (Kumaran et al., 2016) neural architecture. Considering the similarity between the cross-domain transfer and the way human acquire new skills from prior knowledge, NeuroAI holds great potential to overcome effective cross-domain knowledge transfer. However, given various learning behaviors in human brain, how to couple neuroscience with spatiotemporal network tailored for effective transfer and evolution is still challenging as following aspects, 1) As the collective intelligence cannot be accomplished in an action, how to progressively learn tasks from different domains from easy to difficult, 2) How to imitate human learning process to disentangle commonality and ensure common container elastic with continually receiving information. 3) How to ensure the common-individual models to quickly accessible for few-shot generalization.

Actually, classical neuroscience theories and new advancement reveal that, 1) synapse is an important structure connecting neurons, which takes the role of message sharing and cooperation(Van de Ven et al., 2020; Kennedy, 2016; Benfenati, 2007), 2) neuron and synapse are activated by neurotransmitters and action potentials, which is analogous to gradients in artificial neural networks (ANNs), where larger gradient intensity implies larger inconsistency between solidified knowledge and new information (Gulledge et al., 2005; Hussain & Al Alili, 2017; Zenke et al., 2017). Inspired by observations, we theoretically analyze data-driven knowledge space within neural network can be increased with commonality via information entropy. To this end, we propose a **Syn**aptic **EVO**lutional spatiotemporal network, SynEVO, to enable easy cross-domain transfer. Our SynEVO couples three neuro-inspired structures. First, to learn the tasks progressively, we devise a curriculum-guided sample group re-ordering to monotone increasing learning difficulty via task-level gradient relations. Second, to disentangle cross-domain commonality, we borrow the cerebral neocortex and hippocampus structures in brain, and design complementary dual learners, an Elastic Common Container with growing capacity as the core synaptic function to receive cross-domain information for elastically collecting new regularities, and a Task independent Personality Extractor to characterize individual task features for adaptation. Finally, an adaptive dynamic coupler with a difference metric is devised to identify whether the new sample group should be incorporated into common container for enabling model evolution and increasing collective intelligence, which reduces the pollution of inappropriate data. The contributions are three-fold.

- Inspired by neuroscience, we theoretically analyze the increased generalization capacity with cross-domain intelligence and the facilitated task convergence from progressive learning, and a novel NeuroAI framework is proposed to tackle the cross-domain challenge and empower model evolution.

- By human-machine analogy, we introduce an ST-synapse, and couple curriculum and complementary learning with synapse to realize the progressive learning and commonality disentanglement. The model evolution is achieved by elastic common container growing and adaptive sample incorporation.

- Extensive experiments show the collective intelligence increases the model generalization capacity under both source and temporal shifts by at 0.5% to 42%, including few-shot and zero-shot transfer, and empirically validate the convergency of progressive curriculum learning. The extremely reduced memory cost, i.e., only 21.75% memory cost against SOTA on iterative model training and evolution advances urban computing towards sustainable computing paradigm.

## 2. Related Works

**Spatiotemporal learning and its generalization capacity.** Spatiotemporal learning has been investigated to enable convenient urban life (Zhang et al., 2017; 2020; Ye et al., 2019; Zhou et al., 2020; Wu et al., 2019; Liu et al., 2024; 2025a;b; Miao et al., 2024; 2025). However, with urban expansion and economic development, it increases the concerns of distribution shifts as all previous learning models assume independent identical distribution. CauSTG devises causal spatiotemporal learning to explicitly address the temporal domain shifts (Zhou et al., 2023). Concurrently, continuous spatiotemporal learning becomes a prevalent topic for model update to counter temporal shifts (Wang et al., 2023a; Chen et al., 2021). Although prosperity, researchers find that data from different sources in a same system tend to share some common patterns. Some pioneering literature exploits unified multi-task spatiotemporal learning to accommodate diverse modalities where a prompt-empowered universal model is proposed (Yuan et al., 2024), while Yi, et,al. investigates the task-level continuous learning to iteratively capture the common and individual patterns (Yi et al., 2024). Even though, these models still ignore three important issues. 1) Fail to unify the source domain and temporal domain for transfer in a same learning architecture where commonality and individuality are well-decoupled. 2) In a same system, how to quantify relations among tasks and filtering the related ones to facilitate commonality is not clear. 3) Previous models fail to evolve with distribution changes to realize efficient cross-domain few-shot generalization. In contrast, we overcome the cross-domain adaptation through lens of NeuroAI by imitating human

acquiring new knowledge and skills, to achieve evolutional spatiotemporal network.

**Neuro-inspired Artificial Intelligence (NeuroAI).** NeuroAI is an emerging research topic which designs ANNs from the inspiration of neuroscience and the way how human acquire new knowledge (Luk & Christodoulou, 2024). ANN is originated from the biological neurons and researchers teach machine learners to learn like humans (Lindsay, 2020; Zhang & Zhang, 2018). The literature can be classified as two folds from macroscopic to micro-structures. In a macro perspective, human brain usually progressively acquires the knowledge from easy to difficult, which inspires curriculum learning in machine intelligence (Wang et al., 2021; Blessing et al., 2024), and complementary learning scheme with hippocampus and neocortex structures (Kumaran et al., 2016; O'Reilly et al., 2014; Arani et al., 2022). To provide feedback on AI model, reinforcement learning is proposed, and reinforcement learning from human feedback (RLHF) is incorporated into LLMs for thinking like human (Lee et al., 2023). On the micro aspect, brain neuron is activated by neurotransmitters and action potentials with a threshold for activation, where the message passing occurs when there is large potential difference (Zhang & Zhang, 2018). Analogously, gradients in ANN are similar to potential difference in brain neurons, and gradient can be viewed as the knowledge gap between new data and trained models, thus gradients can be exploited to interpret the relation between model and sample groups. Synapse is also an essential structure for bridging the message between neurons, where pioneering researches have demonstrated the potential of knowledge transfer by imitating synapse structure (Zenke et al., 2017; Hussain & Al Alili, 2017). Actually, unveiling the relations between brain structures and AI model transfer mechanism can advance the model evolution. Despite prosperity, on cross-domain transfer in spatiotemporal learning, how to investigate the specific mechanism that adapting to brain learning on transfer and generalization is still under-explored.

## 3. Preliminaries

**Spatiotemporal Cross-Domain Observations.** In an urban system, data can be collected from different sources. We can model diverse spatiotemporal learning tasks as spatial-temporal graph prediction, and the deterministic observations can be defined as $(x_i^j)_c$, which is an element in $\mathbb{X} = \{X_1, X_2, ..., X_C\} \in \mathbb{R}^{N \times T \times C}$. The $(x_i^j)_c$ indicates the value on graph node $j$ at timestamp $i$ from $c$-th data source, where $C$ represents the number of sources. As the data distribution can be shifted across temporal steps, then the task domain can be classified into both different temporal domains with changed distribution and source domains.

**Neuro-Inspired Cross-Domain Learning.** We define neuro-inspired cross-domain spatiotemporal learning model as an evolution model $\mathcal{M}$, i.e.,

$$\widehat{Y} = \mathcal{M}(X_1, X_2, \ldots, X_k; \theta_{\mathcal{M}}) \qquad (1)$$

where $\theta_{\mathcal{M}}$ denotes the learnable parameters of model $\mathcal{M}$. When the data from new domain $X_{k+1}$ comes, we aim to quickly adapt $\mathcal{M}$ to $\mathcal{M}'$, i.e.,

$$\mathcal{M}' \leftarrow \mathcal{M}(X_{k+1}, \theta_{\mathcal{M}}; \theta_{\mathcal{M}'}) \qquad (2)$$

where $\theta_{\mathcal{M}'}$ denotes learnable parameters of updated $\mathcal{M}'$.

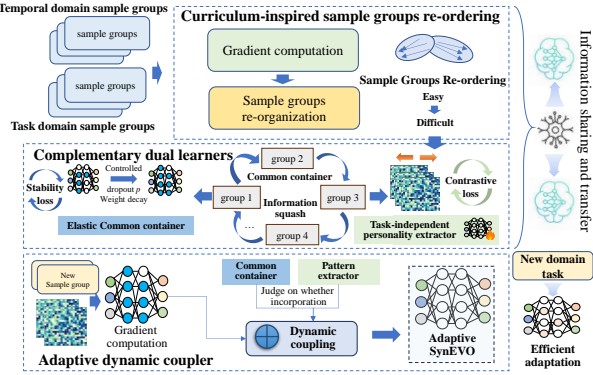

*Figure 1.* Framework Overview of SynEVO

**Proposition 3.1.** *Increased information with cross-domain learning. Given spatiotemporal data observations from different sources $\{X_1, X_2, X_3, ..., X_k\}$ and then there must be shared commonality among domain data patterns, i.e.,*

$$\forall i, j(1 \le i < j \le k), I(X_i; X_j) > 0 \qquad (3)$$

*then the well-learned information from the cross-domain learning model $\mathcal{M}$ is increased by continually receiving domain knowledge, i.e.,*

$$Info(\mathcal{M}(X_1, ..., X_k; \theta_{\mathcal{M}})) > Info(\mathcal{M}(X_1, ..., X_{k-1}; \theta_{\mathcal{M}})) >$$
$$... > Info(\mathcal{M}(X_1; \theta_{\mathcal{M}}))$$
$$(4)$$

*where Info is the information encapsulated in $\mathcal{M}$.*

The Prop.3.1 delivers that cross-domain learning with different data sources 'in harmony with diversity' can increase the learned information in model $\mathcal{M}$ and it can be proved in Appendix. A.

## 4. Methodology

### 4.1. Framework Overview

SynEVO constructs a neural synaptic spatiotemporal network to share and transfer cross-domain knowledge for

generalization and adaptation. As illustrated in Fig. 1, our synaptic neural structure consists of three components, curriculum-inspired sample group re-ordering to determine the learning order of sequential samples from easy to difficult, complementary dual common-individual learners including an Elastic Common Container and task-independent personality extractor [1] to disentangle the task commonality and personality and an adaptive dynamic coupler to aggregate the dual learners so as to adapt shared patterns and preserve the individuality of tasks. It is noted that the Spatiotemporal learner backbone is implemented by Graph-WaveNet (GWN) (Wu et al., 2019).

## 4.2. Curriculum Guided Task Reordering

Learning from easy to difficult is a common practice for human acquiring knowledge and skills, which is named as curriculum learning (Bengio et al., 2009), e.g., the teaching process in our class also follows inculcating knowledge from basic to improved ones. To capture the commonality of data patterns in different domains, we propose curriculum guided task re-ordering. From an optimization perspective, directly confronting a complex task can lead the model to be caught in poor local optimums with exploding gradients. In contrast, by starting from a simple task and gradually increasing the difficulty, the model can be effectively guided to converge in the direction of global optimum, which enables better exploration in parameter space.

Gradients can characterize the consistency degree between training model and new sample groups, thus the gradients are exploited to indicate the difficulty of adapting models to samples. We then exploit the gradient to compute the adaptation between new feeding samples and the training model, and determine the learning order via imitating the curriculum learning process. Specifically, we apply the backward of loss to compute the gradient. For each input sample group $X_c$ from $c$-th domain, we initial a trainable model $\mathcal{M}_c(X_c; \theta_{\mathcal{M}_c})$ and train $\mathcal{M}_c$ until the loss function of it converges. Then we backward the final loss to compute the gradients of $\theta_{\mathcal{M}}$ as $\{\nabla_1, \nabla_2, \ldots, \nabla_n\}$, where $\nabla_i$ denotes the gradient of the $i$-th layer of $\theta_{\mathcal{M}}$ and $n$ denotes the the number of the layers of $\theta_{\mathcal{M}}$. After that, compute the sum of squares of the gradients by,

$$sum_c = \sum_{i=1}^{n} ||\nabla_i||_2^2 \qquad (5)$$

where $||\nabla_i||_2^2$ denotes the square of the L2 norm of $\nabla_i$. Then, we concatenate the gradient to denote the overall

consistency between data and model by,

$$cat_c = [\nabla_1^{(expand)}||\nabla_2^{(expand)}|| \ldots ||\nabla_n^{(expand)}] \qquad (6)$$

where $\nabla_i^{(expand)}$ denotes the expanded tensor of $\nabla_i$ and $||$ denotes the concatenation of tensors.

With obtaining all $cat_c$ for input data $X_c$, we identify the minimum value among them by $min = \arg\min_c sum_c$, which is considered as the compared bench. Next, to re-order other sample groups, we compute the vector difference between $cat_c$ and the bench one $cat_{min}$ by,

$$d_c = cat_c \ominus cat_{min} \qquad (7)$$

where $\ominus$ is the element-wise minus for vectors. After that, we can reorder the input sample groups $\{X_1, X_2, \ldots, X_k\}$ based on the length of $d_c$, i.e., $l(d_c)$ in ascending order and get the ordered sequence $\mathcal{S} = \{X_{c_1}, X_{c_2}, \ldots, X_{c_k}\}$, where $k$ is the number of sample groups. Therefore, we can capture the inner relation between sample groups by gradients, which avoids the isolation of information, and allows the learning process from easy to difficult.

## 4.3. Complementary Dual Common-individual Learners

Inspired by complementary functions in brain memory, we construct a dual common individual learners to respectively accommodate two major knowledge based on three insights, 1) Complementary memory where neocortex remembers long-term and stable skills while hippocampus acquires new and quick knowledge. 2) More neurons are activated with knowledge increasing. 3) Distinguished patterns makes long-standing memory. Overall, our design inherits the complementary learning scheme into respective common container and personality pattern extractor. The common container is devised accounting for the core synaptic function to receive cross-domain common information with elastically increasing collective intelligence, and a task independent personality extractor is to characterize individual task features for quick adaptation. They cooperate with each other for generalized cross-domain learning.

### 4.3.1. ELASTIC COMMON CONTAINER

Deep learning models iteratively trained with new samples can automatically fuse patterns across all samples. Based on above analysis, the commonality should be expanded when the acquired knowledge is increasing with iteratively feeding into new samples. With well-organized task sequences, we are expected to mimic such knowledge expansion in brains for neural networks. In detail, we borrow a couple of simple yet effective strategies in deep learning to empower the model with elastic property. *Dropout* and L2 Regularization with *weight decay* control the overall complexity of model that potentially avoid over-fitting by the number of

---

[1] As observations in source and temporal domains are organized into sample groups, which can be viewed as various tasks, here we interchangeably utilize sample group and tasks in our main text.

active neurons. For Dropout, every neuron can be set as zero (dropout) with probability $p$ and each weight decay coefficient weight $\lambda$ for L2 controls the importance of L2 item. The smaller probability $p$ and smaller weight decay $\lambda$, the model is more active. Despite promising, how to quantitatively control the activeness of neurons by probability $p$ and weight decay $\lambda$ is still unclear. To address

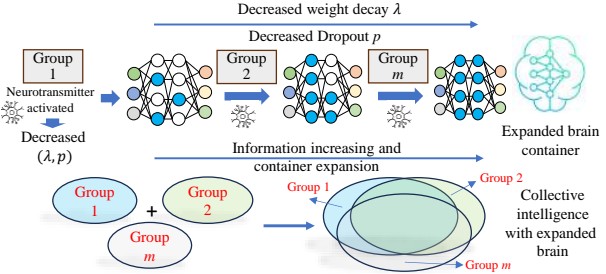

*Figure 2.* The process of elastic growth of common container

such quantitation challenge, we introduce Lemma. 4.1 from neuroscience (Gulledge et al., 2005).

**Lemma 4.1.** *The probability of presynaptic neurotransmitter release can be described by a propagation model(Bertram et al., 1996; Schneggenburger & Neher, 2000),*

$$P_r = P_0(1 - e^{-\tau}) \qquad (8)$$

where $P_r$ denotes the probability of neurotransmitter release, $P_0$ denotes the basic release probability, $\tau$ is the successive activeness difference between a pre-synaptic neuron and after-synaptic neuron.

Lemma. 4.1 suggests that the neurotransmitter can induce the activeness of neurons and we can exploit the such electric potential difference to mimic the process. Fortunately, earlier we have discussed that gradient can indicate the consistency between model and new samples, thus we take the second-order difference, the successive gradient variation $|\boldsymbol{d}|$ based on Eq. 7 as the propagation degree $\tau$.

For dropout, we first define the matrix of all learnable parameter in model $\mathcal{M}$ as $\boldsymbol{M}$, and provide the following definition of activated model parameters.

**Definition 4.2.** Activated model parameters. *For parameter matrix* $[\boldsymbol{M}]$*, its activeness matrix is defined as,*

$$[\boldsymbol{A}]^{x,y} = \begin{cases} 0 & if\ [\boldsymbol{M}]^{x,y}\ is\ dropped\ out, \\ 1 & otherwise. \end{cases} \qquad (9)$$

where $[\boldsymbol{A}]^{x,y}$, $[\boldsymbol{M}]^{x,y}$ denotes the element of matrix $[\boldsymbol{A}]$, $[\boldsymbol{M}]$ on position $(x, y)$ respectively. Based on the activeness matrix, the activated model parameters can be updated as, $\boldsymbol{M} = \boldsymbol{M} \odot \boldsymbol{A}$, where $\odot$ is the Hadamard product.

The larger number of non-zero elements in $[\boldsymbol{A}]$ indicates more model activeness and capture more information with decreased probability $p$.

With above Lemma. 4.1, we further model the dropout factor $p_c$ as,

$$p_c(\boldsymbol{d}_c) = p_0(1 - e^{l(\boldsymbol{d}_c)-d_{max}})(0 < p_0 \leq 1) \qquad (10)$$

where $p_c$ denotes the dropout factor for domain $c$, $p_0$ is a hyperparameter, $\boldsymbol{d}_c$ denotes the vector difference for domain $c$ against $\boldsymbol{cat}_{min}$ and $d_{max}$ is the maximum length of all gradient vectors $\boldsymbol{d}_c$.

Similarly, for the *weight decay* coefficient of L2 regularization that controls the model in the optimizer, we can re-write the dynamic update of weight decay according the variation of gradient as below,

$$\lambda_c(\boldsymbol{d}_c) = \lambda_0(1 - e^{l(\boldsymbol{d}_c)-d_{max}})(0 < \lambda_0 < 1) \qquad (11)$$

In Eq. 11, $\lambda_c$ decreases as $l(\boldsymbol{d}_c)$ increases, which realize the elastic growth of the model and improves the generalization of the model. To this end, we consider that controlling the dropout value $p_c$ and weight decay coefficient $\lambda_c$ with the vector $\boldsymbol{d}_c$ can efficiently realize the gradual release of model activeness, which has been shown in Fig. 2. With our synapse structure, common patterns are iteratively enhanced. As new domain arrives, the overall knowledge boundary and learned parameter space of our common container are expanding. Then can do quick and light-weight adaptation from the existing knowledge space to new domain.

### 4.3.2. TASK INDEPENDENT PERSONALITY EXTRACTOR

Besides commonality, the personalized pattern of each task especially new task is also vital for domain adaptation. Formally, it is expected to derive an additional personality extractor $g$, which transforms the input $\boldsymbol{X}_c$ to the output $\boldsymbol{E}_c$, i.e., $\boldsymbol{E}_c = g(\boldsymbol{X}_c; \boldsymbol{W}_g) = \boldsymbol{W}_g \boldsymbol{X}_c$. Here, we define a new criterion $\mathcal{D}(\boldsymbol{A}, \boldsymbol{B})$ to measure the difference between tensors $\boldsymbol{A}$ and $\boldsymbol{B}$, i.e.,

$$\mathcal{D}(\boldsymbol{A}, \boldsymbol{B}) = \sqrt{(\boldsymbol{A} \ominus \boldsymbol{B})^2} \qquad (12)$$

Inspired by distinguishing patterns in memory for activating the neuron activeness, we explore the contrastive learning (Khosla et al., 2020; Hadsell et al., 2006) to implement such personality extractor,

$$\begin{aligned} \mathcal{R}(\boldsymbol{E}_i, \boldsymbol{E}_j; \boldsymbol{W}_g) = &\hat{y}\mathcal{D}(\boldsymbol{E}_i, \boldsymbol{E}_j) \\ &+ (1 - \hat{y})max(0, m - \mathcal{D}(\boldsymbol{E}_i, \boldsymbol{E}_j)) \end{aligned} \qquad (13)$$

where $\mathcal{R}(\boldsymbol{E}_i, \boldsymbol{E}_j)$ refers to contrastive objective function between two representations $\boldsymbol{E}_i, \boldsymbol{E}_j$. The domain indicator $\hat{y} = 1$ if $\boldsymbol{X}_i$ and $\boldsymbol{X}_j$ come from the same domain while $\hat{y} = 0$ indicates two samples from different domains, and $m$

controls the minimum distance between $X_i$ and $X_j$ from different domains. Since the personality extractor $g$ updates to approach the minimization objective, the representations $E_i$ and $E_j$ from the same domain become closer and the representations $E_i$ and $E_j$ from different domains are away from each other.

Therefore, the commonality and personality can be disentangled with the contrastive learning objective. As samples of a new domain comes, the personality extractor is capable of extracting the representation of the new domain, i.e., $E_c = g(X_c; W_g)$, and the commonality patterns are learned by the common container for commonality growth, which significantly enhances the ability of the model to comprehend both previous and new knowledge.

### 4.4. Adaptive Dynamic Coupler

To achieve cross-domain adaptation while maintaining both personality and commonality, we construct an adaptive dynamic coupler to aggregate the commonality and personality. When the sample group from a new domain $X_{k+1}$ comes, the task independent personality extractor first extracts the representation of $X_{k+1}$ by $E_{k+1} = g(X_{k+1}; W_g)$. Then we preserve a list $G$ which contains the distance between the representation of the new domain and the trained domains based on Eq. 12. To be specific, $G$ is defined as, $G = \{\mathcal{D}_1, \mathcal{D}_2, \ldots, \mathcal{D}_k\}$, where $k$ is the number of trained domains in the elastic common container and $\mathcal{D}_i$ denotes $\mathcal{D}(E_{k+1}, E_i)$. Let $\mathcal{D}_{min}$ be the minimum $\mathcal{D}_i$ in $G$ and if $0 < \mathcal{D}_{min} < \kappa$ (where $\kappa$ is a threshold which is a hyperparameter), it indicates that the new domain shares potential commonality with the trained domains. Then we can put it into the common container to compute the gradient and dynamically adjust the dropout factor $p_{k+1}$ and the weight decay coefficient $\lambda_{k+1}$. At last, the common container trains the new domain based upon the adjusted factors and thus absorb the knowledge from the new domain to realize elastic growth. If $\mathcal{D}_{min} \geq \kappa$, which indicates that the new domain almost shares no commonality with trained domains, then we re-instantiate personality extractor by initializing learnable parameters with previous extractor for a quick adaptation. The comparison of representations is implemented by a gate structure $h$,

$$h(\mathcal{D}_{min}, \kappa) = \begin{cases} 1 & if \ 0 < \mathcal{D}_{min} < \kappa \\ 0 & otherwise. \end{cases} \quad (14)$$

To conclude, the overall learning objective of our model $\mathcal{M}'$ to the input $X_{k+1}$ can be defined as,

$$\begin{aligned} Loss(\theta_{\mathcal{M}'}) = &h(\mathcal{D}_{min}, \kappa)(L(\mathcal{M}'(X_{k+1}, \theta_{\mathcal{M}}; \theta_{\mathcal{M}'}), Y_{k+1}) \\ &+ \lambda_{k+1}||\theta_{\mathcal{M}'}||_2^2) \\ &+ (1 - h(\mathcal{D}_{min}, \kappa))L(\mathcal{M}'(X_{k+1}, \theta_{init}; \theta_{\mathcal{M}'}), Y_{k+1}) \end{aligned}$$
$$(15)$$

where $Y_{k+1}$ denotes the target value of $\mathcal{M}'(X_{k+1})$, $\lambda_{k+1}$ is computed based on Eq. 10, $||\theta_{\mathcal{M}'}||_2^2$ denotes the square of

L2 norm of $\theta_{\mathcal{M}'}$ and $\theta_{init}$ denotes the new random initial learnable parameters of $\mathcal{M}'$.

## 5. Experiment

### 5.1. Datasets

We collect and process four datasets for our experiments: 1) **NYC** (NewYorkCity, 2016): Include three months of traffic data consisting of four source domains which are CrowdIn, CrowdOut, TaxiPick and TaxiDrop collected from Manhattan in New York City. 2) **CHI** (CHICAGO, 2023): Consist of three source domains of traffic status, which are Risk, TaxiPick and TaxiDrop collected in the second half of 2023 from Chicago. 3) **SIP**: Includes three months of traffic data consisting of two source domains which are Flow and Speed collected from Suzhou Industrial Park. 4) **SD** (Liu et al., 2023): Include traffic flow data collected from San Diego in 2019.

### 5.2. Evaluation Metrics and Baselines

We apply three evaluation metrics in our experiments, which are mean absolute error (MAE), root mean square error (RMSE) and mean absolute percentage error (MAPE). We exploit seven prevalent baselines for evaluations, including STGNNs (**STGCN** (Yu et al., 2017), **STGODE** (Fang et al., 2021), **GWN** (Wu et al., 2019)), RNN-based models (**AGCRN** (Bai et al., 2020)) and attention-based models (**STTN** (Xu et al., 2020), **ASTGCN** (Guo et al., 2019), **CMuST** (Yi et al., 2024)).

### 5.3. Implementation Details

We split the datasets into training, validation and testing sets with the ratio of 7:1:2. Datasets NYC, CHI and SIP are utilized for validation on cross-source and cross-temporal domain tasks, while SD with one attribute but large time span is for cross-temporal domain evaluation. For the first three datasets, on cross-domain evaluation, we leave TaxiPick, TaxiDrop and Speed as the evaluation domain on respective NYC/CHI/SIP sets, and let other data sources to be trained iteratively. For their cross-temporal domain evaluation, we divide one day into four equal periods, and leave the last period of a day on all source domains for evaluation. Regarding cross-temporal domain validation on SD, we divide one day into six equal periods and leave the last period for evaluation of temporal domain adaptation. We run each baseline three times and report the averaged results to reduce the influence of randomness issue. For curriculum-guided task reordering, Adam optimizer (Kingma, 2014) is applied with initialized learning rate of 0.01 and weight decay of 0.001 for the initial learnable model $\mathcal{M}_c$. For complementary dual learners, we use the mean square error (MSE) as the criterion $\mathcal{D}$ of the personality extractor. For

*Table 1.* Performance comparison on cross-source adaptation on NYC, CHI, SIP and cross-temporal adaptation on SD. Best results are **bold** and the second best results are underlined.

| METHODS | NYC | | | CHI | | | SIP | | | SD | | |
|---|---|---|---|---|---|---|---|---|---|---|---|---|
| | MAE | RMSE | MAPE | MAE | RMSE | MAPE | MAE | RMSE | MAPE | MAE | RMSE | MAPE |
| STGCN | 6.774 | 15.853 | 0.374 | 1.518 | 2.804 | 0.415 | 0.753 | 1.492 | 0.219 | 12.477 | 22.055 | 0.182 |
| STGODE | 9.522 | 22.555 | 0.481 | 1.543 | 2.792 | 0.433 | 0.732 | 1.460 | 0.211 | 12.300 | 20.808 | 0.180 |
| GWN | 10.263 | 24.535 | 0.546 | 1.520 | 2.873 | 0.421 | 0.737 | 1.473 | 0.212 | 18.890 | 29.220 | 0.230 |
| STTN | 7.962 | 19.544 | 0.435 | 1.494 | **2.699** | 0.409 | 0.732 | 1.461 | 0.212 | 13.092 | 22.054 | 0.183 |
| AGCRN | 8.254 | 19.301 | 0.488 | 1.543 | 2.805 | 0.427 | 0.743 | 1.469 | 0.215 | 12.225 | 22.094 | 0.179 |
| ASTGCN | 10.323 | 25.070 | 0.519 | 1.536 | 2.809 | 0.413 | 0.743 | 1.473 | 0.215 | 13.079 | 22.047 | 0.185 |
| CMuST | 6.576 | 14.954 | 0.459 | 1.498 | 2.781 | 0.388 | 0.737 | 1.497 | 0.219 | **10.940** | 19.113 | 0.162 |
| SYNEVO | **6.494** | **14.885** | **0.358** | **1.486** | 2.733 | **0.361** | **0.697** | **1.390** | **0.205** | 10.984 | **18.654** | **0.157** |

*Table 2.* GPU cost comparison between CMuST and SynEVO

| METHODS | GPU COST | | | |
|---|---|---|---|---|
| | NYC | CHI | SIP | SD |
| CMuST | 4034MB | 4118MB | 2450MB | 19533MB |
| SYNEVO | 2170MB | 2210MB | 2044MB | 4252MB |

*Table 3.* Detailed comparison on cross-temporal adaptation and cross-source adaptation on NYC, CHI and SIP (cross-source results are from Tab. 1)

| | | NYC | CHI | SIP |
|---|---|---|---|---|
| TEMPORAL | SYNEVO MAE | 6.278 | 1.457 | 0.683 |
| | CMuST MAE | 6.457 | 1.472 | 0.716 |
| SOURCE | SYNEVO MAE | 6.494 | 1.486 | 0.697 |
| | CMuST MAE | 6.576 | 1.498 | 0.737 |

Elastic Common Container, the loss criterion is adopted with widely-used MaskedMAELoss. We run STGODE, STTN, CMuST on SD on NVIDIA A100-PCIE-40GB and other experiments on Tesla V100-PCIE-16GB by adapting the model scale with GPU versions.

## 5.4. Performance Comparison

1) **Comparison among baselines.** Comparison results can be found in Tab. 1, which reports results of cross-source domain adaptation on the NYC, CHI, SIP and results of cross-temporal domain adaptation on the SD dataset. In general, our SynEVO model outperforms other baselines across most metrics on four datasets. Compared with those without a commonality extraction mechanism, the overall adaptation performance of CMuST and SynEVO with commonality significantly outperforms them, where SynEVO outperforms other baselines except CMuST on average by about 25.0% on NYC, 2.6% on CHI, 5.8% on on SIP and 18.3% on SD. These results indicate the effectiveness of our elastic common container, which allows our model to dynamically grow in an appropriate area and capture the inner relation of different domains. At the same time, our SynEVO outperforms the vanilla backbone GWN by about 35.6% on average of four datasets. It's worth noting that on NYC and SD, GWN seems to be trapped in a local optimum, and it further demonstrates that the adaptive structure enhances the model's transfer and adaptation. Although attention based model CMuST achieves satisfactory performance, CMuST exactly costs much more computing resource than our SynEVO as shown in Tab. 2, especially the 4.59 times of computation cost of SynEVO on large dataset SD. It shows that the the NeuroAI structure can enable lightweight adaptation which ensures superior performance with

less computing resource. 2) **Detailed comparison on cross-temporal domain adaptation and cross-source domain adaptation against SOTA.** We evaluate the performance of cross-temporal domain adaptation and cross-source domain adaptation of SynEVO on NYC, CHI and SIP against a selected best baseline CMuST, as shown in Tab. 3. Obviously, our SynEVO can outperform CMuST on almost three datasets on MAE, and the improvement on source-domain is more significant than temporal domain, which empirically verifies our NeuroAI-based synapse solution captures the complex commonality across domains and expand the boundary of learning space.

## 5.5. Ablation Study

In order to uncover the significance of each module to the success of SynEVO, we perform an ablation study on cross-temporal domain adaptation via removing each module on the four datasets. The ablated variants are as follows. **1) SynEVO-REO:** Remove the module of curriculum-guided sample group reordering. **2) SynEVO-Ela:** Remove the dynamic adjustment of dropout factor $p$ and weight decay coefficient $\lambda$ with a static value, e.g., $p = 0.1, \lambda = 0.001$. **3) SynEVO-PE:** Without relation comparison on individual and commonality, any domain even unrelated sample groups can be fed into the model without a judgment gate.

Tab.4 shows the results of ablation studies. In general, removing abovementioned modules leads to the consistent performance drops. When removing the module of elastic growth, the performance drops the most by 44.2% on MAE, indicating the most important structure of elastic com-

*Table 4.* Ablation studies of SynEVO doing cross-temporal domain adaptation on four datasets

| METHODS | NYC | | | CHI | | | SIP | | | SD | | |
|---|---|---|---|---|---|---|---|---|---|---|---|---|
| | MAE | RMSE | MAPE | MAE | RMSE | MAPE | MAE | RMSE | MAPE | MAE | RMSE | MAPE |
| SYNEVO | 6.278 | 14.570 | 0.357 | 1.457 | 2.696 | 0.359 | 0.683 | 1.369 | 0.193 | 10.984 | 18.654 | 0.157 |
| SYNEVO-REO | 7.032 | 17.228 | 0.383 | 1.785 | 3.210 | 0.496 | 0.712 | 1.400 | 0.214 | 14.604 | 22.669 | 0.166 |
| SYNEVO-ELA | 8.384 | 19.142 | 0.386 | 1.907 | 3.414 | 0.471 | 0.745 | 1.418 | 0.223 | 17.481 | 27.204 | 0.245 |
| SYNEVO-PE | 7.451 | 16.988 | 0.382 | 1.711 | 3.098 | 0.479 | 0.732 | 1.444 | 0.212 | 16.382 | 24.276 | 0.187 |

mon container for model evolution and adaptation. More specifically, when removing sample re-ordering, the performance drops by about 18.1%, while personality extractor is discarded, the performance drops by about 19.1%, which shows the personality and re-instantiate mechanism is also critical for ensuring model robustness and consistency.

### 5.6. Detailed Analysis

**Uncovered sample group sequences for curriculum learning.** In curriculum guided task reordering, the reordered the input group samples are illustrated in Fig. 3(a) based on the gradients. In our experiments, we find that domains from the same source are not necessarily next to each other in the ordered sequence $\mathcal{S}$, which verifies that SynEVO has successfully uncovered hidden correlation information between domains. Moreover, it is demonstrated that reordering our input sample groups can learn certain commonality, empirically providing evidence for the effectiveness of cross-domain learning.

**Observed quick adaptation via loss behavior.** In elastic common container, the sample groups are periodically fed into models. We let SynEVO elastically grow to absorb the commonality by iteratively feeding sample groups, and visualize the training loss of two consecutive learning cycles on SD in Fig. 3(b). The blue curve denotes the first training cycle while the red one denotes the second. These two cycles share the same training data and order. Every mutation in the curve represents the input of a new domain. Obviously, the loss of the second cycle is much lower than the former one and the loss drops quickly after new domain input, which shows the quick adaptation by constructing learning tasks in a cycled and elastic manner.

**Effective zero-shot adaptation.** To further evaluate the adaptation performance of SynEVO, we conduct zero-shot cross-temporal domain adaptation, i.e., testing without training, and make comparisons with the backbone GWN, where results are shown in Tab.5. Obviously, SynEVO outperforms better than GWN by averagely 29.8% on the four datasets, and it numerically proves that SynEVO has captured the hidden commonality of various input data so that it can achieve effective and even superior performance on zero-shot tasks.

**More empirical analysis on task reordering and design of commonality extraction.** We supplement the experi-

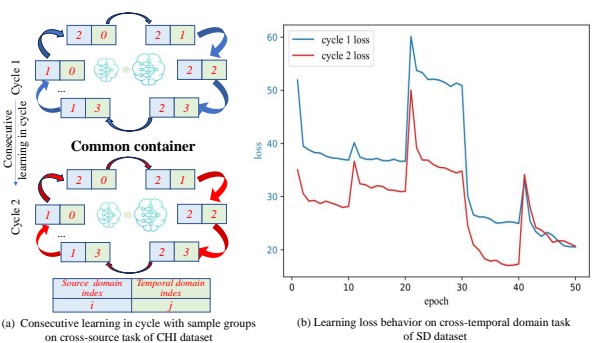

(a) Consecutive learning in cycle with sample groups on cross-source task of CHI dataset

(b) Learning loss behavior on cross-temporal domain task of SD dataset

*Figure 3.* Training order on CHI and training loss behavior on SD

*Table 5.* Comparison of zero-shot cross-temporal domain adaptation performance

| | | NYC | CHI | SIP | SD |
|---|---|---|---|---|---|
| SYNEVO | MAE | 13.420 | 1.995 | 0.775 | 16.981 |
| | RMSE | 32.059 | 3.769 | 1.510 | 25.197 |
| | MAPE | 0.668 | 0.369 | 0.217 | 0.214 |
| GWN | MAE | 17.091 | 4.109 | 1.039 | 21.446 |
| | RMSE | 40.802 | 8.385 | 2.056 | 30.736 |
| | MAPE | 0.856 | 0.742 | 0.264 | 0.271 |

ments of **1) H2E:** reordering tasks from **hard to easy** to emphasize the importance of training order. **2) SynEVO-IL:** eliminating iterative learning for commonality, i.e., only training and testing on one dataset. **3) DER:** setting a different neural expansion rate $p_c$, e.g., $p_c(\boldsymbol{d}_c) = p_0/l(\boldsymbol{d}_c)$. Results are shown in Tab.6. The reversed order falls into inferior performances, which emphasizes the significance of reordering the tasks from easy to difficult in our design. Moreover, hard-to-easy performances are better than random ordering of SynEVO-REO, which may be attributed to common relations between neighboring tasks as they are ordered even the reverse one. Based on the experimental results, we can conclude our commonality learner and the setting of p are reasonable and empirically justified.

### 5.7. Hyperparameter Sensitivity Analysis

We varied Dropout $p_0$ from $\{0.1, 0.3, 0.5, 0.7, 1\}$, weight decay coefficient $\lambda_0$ from $\{0.01, 0.03, 0.05, 0.07, 0.1\}$, and distance threshold $\kappa$ from $\{1 \times 10^3, 1 \times 10^4, 1 \times 10^5, 1 \times 10^6\}$. Results shown in Fig.4-Fig.7 indicate that the optimal settings are $\kappa = 1 \times 10^3$ on all datasets, $p_0 =$

*Table 6.* More empirical analysis on task reordering and design of commonality extraction

| METHODS | NYC | | | CHI | | | SIP | | | SD | | |
|---|---|---|---|---|---|---|---|---|---|---|---|---|
| | MAE | RMSE | MAPE | MAE | RMSE | MAPE | MAE | RMSE | MAPE | MAE | RMSE | MAPE |
| H2E | 7.217 | 18.807 | 0.415 | 1.632 | 3.066 | 0.405 | 0.705 | 1.394 | 0.208 | 11.636 | 19.789 | 0.163 |
| SYNEVO-IL | 8.201 | 19.090 | 0.423 | 1.554 | 2.887 | 0.385 | 0.711 | 1.412 | 0.216 | 13.128 | 20.890 | 0.220 |
| DER | 7.213 | 16.310 | 0.422 | 1.550 | 2.765 | 0.367 | 0.705 | 1.399 | 0.207 | 11.944 | 19.599 | 0.168 |

$0.5, \lambda_0 = 0.05$ on NYC and SIP, $p_0 = 1, \lambda_0 = 0.1$ on CHI and $p_0 = 0.7, \lambda_0 = 0.07$ on SD. For $\kappa$, if the threshold is extremely small, it means no new domain is allowed in, which will violate the principle of 'harmony with diversity for collective intelligence' then we can observe that with increasing and more relaxed condition for fusion, more noise will be introduced to reduce the performances. Thus the trade-off between commonality and individual feature extraction should be obtained during model design.

# 6. Conclusion

In this paper, we propose a novel NeuroAI framework SynEVO to enable cross-domain spatiotemporal learning for few-shot domain adaptation. From neuroscience theories, a curriculum-guided sample group re-ordering, a couple of complementary dual learners which includes an elastic common container, and a task independent personality extractor are proposed to capture commonality in an elastic manner. The adaptive dynamic coupler determines whether the new feeding samples can be aggregated into SynEVO for achieving model evolution. Extensive experiments on both cross-source and cross-temporal domains validate the 0.5% to 42% improvements against baselines. For future work, we plan to mine the more inner mechanism of human brain to facilitate the generalization of general AI models. Moreover, our model can be generally nested within other neural networks. In other areas, it is applicable to utilize our evolvable 'data-model' collaboration to decouple the invariant and variable patterns, and reconstruct the OOD distribution with new patterns.

# Acknowledgements

This work was supported by the National Natural Science Foundation of China (No.12227901), Natural Science Foundation of Jiangsu Province (BK.20240460, BK.20240461), the grant from State Key Laboratory of Resources and Environmental Information System. The AI-driven experiments, simulations and model training were performed on the robotic AI-Scientist platform of Chinese Academy of Science.

# Impact Statement

Our work focuses on the research on the aggregation of data in different fields in cities, which has strong practical significance for the deployment of various data-driven urban applications and the improvement of the convenience of urban life. Although our work is inspired by Neuroscience, the theories and data we use are publicly available and do not collect information from any human, animal, or private individuals, so there are no ethical issues.

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

In the Appendix, we provide the necessary proof our proposition and supplementary experiments for model evaluation.

## A. Proof of Proposition 3.1

Cross-domain learning can increase the learned information.

**Proposition A.1.** *Increased information with cross-domain learning. Given spatiotemporal data observations from different sources* $\{X_1, X_2, X_3, ..., X_k\}$ *and the there must share commonality among domain data patterns, i.e.,*

$$\forall i,j (1 \leq i < j \leq k), I(X_i; X_j) > 0 \tag{16}$$

*then the well-learned information from the cross-domain learning model* $\mathcal{M}$ *is increased by continually receiving domain knowledge, i.e.,*

$$Info(\mathcal{M}(X_1, ..., X_k; \boldsymbol{\theta}_{\mathcal{M}})) > Info(\mathcal{M}(X_1, ..., X_{k-1}; \boldsymbol{\theta}_{\mathcal{M}})) > ... > Info(\mathcal{M}(X_1; \boldsymbol{\theta}_{\mathcal{M}})) \tag{17}$$

*where* $Info$ *is the knowledge information encapsulated in model* $\mathcal{M}$.

*Proof.* Here we provide the proof from the perspective of information theory. We first define the mutual information $I$ as,

$$\begin{aligned} I(X; Y) =& H(X) - H(X|Y) = H(Y) - H(Y|X) \\ =& H(X) + H(Y) - H(X, Y) \end{aligned} \tag{18}$$

where $H$ denotes the information entropy.

In our learning scheme, by denoting the input observations $X$ and predicted output $\widehat{Y}$, we transfer the learning objective of MAE into the following equation via the theory of Information Bottleneck (Tishby et al., 2000),

$$L(X, \hat{Y}, Z) = I(X; Z) - \beta I(Z; \hat{Y}) \tag{19}$$

where $\beta$ is a hyperparameter controlling the balance between the mutual information $I(X, Z)$ and $I(Z, \hat{Y})$. Minimizing $I(X; Z)$ aims to reduce the information redundancy of input $X$ while maximizing $I(Z; \hat{Y})$ aims to make sure the representation $Z$ keeps enough information of output $\hat{Y}$.

Considering the relationship between information entropy and mutual information, we have,

$$I(X; Y) = H(X) - H(X|Y) \tag{20}$$

we can transform above Eq.20 into,

$$H(X|Y) = H(X) - I(X; Y) \tag{21}$$

Then, $\forall i (1 \leq i < k)$, we can compute $H(X_i|X_1, X_2, \ldots, X_{i-1})$ as,

$$H(X_i|X_1, X_2, \ldots, X_{i-1}) = H(X_i) - I(X_i; X_1, X_2, \ldots, X_{i-1}) \tag{22}$$

Similarly,

$$H(X_{i+1}|X_1, X_2, \ldots, X_i) = H(X_{i+1}) - I(X_{i+1}; X_1, X_2, \ldots, X_i) \tag{23}$$

Actually, for any two datasets, the initial learning uncertainties of corresponding datasets are equilibrated, and denoted as,

$$\forall i,j (1 \leq i < j \leq k), H(X_i) = H(X_j) \tag{24}$$

Then we can continue our derivation into,

$$H(X_i) = H(X_{i+1}) \tag{25}$$

Then we have assumed the involved datasets are sharing common patterns, with $\forall i,j (1 \leq i < j \leq k), I(X_i; X_j) > 0$, we can obtain,

$$I(X_i; X_1, X_2, \ldots, X_{i-1}) < I(X_{i+1}; X_1, X_2, \ldots, X_i) \tag{26}$$

Therefore, based on Eq.22, Eq.23, Eq.25, Eq.26, we can conclude that,

$$H(X_i|X_1, X_2, \ldots, X_{i-1}) > H(X_{i+1}|X_1, X_2, \ldots, X_i) \tag{27}$$

which means the model is expanding the boundary of learning. Moreover, according to Eq.20, we can further derive that,

$$H(X_i|X_j) < H(X_i) \tag{28}$$

which means the commonality between input data contributes to reducing the uncertainty of the tasks.  □

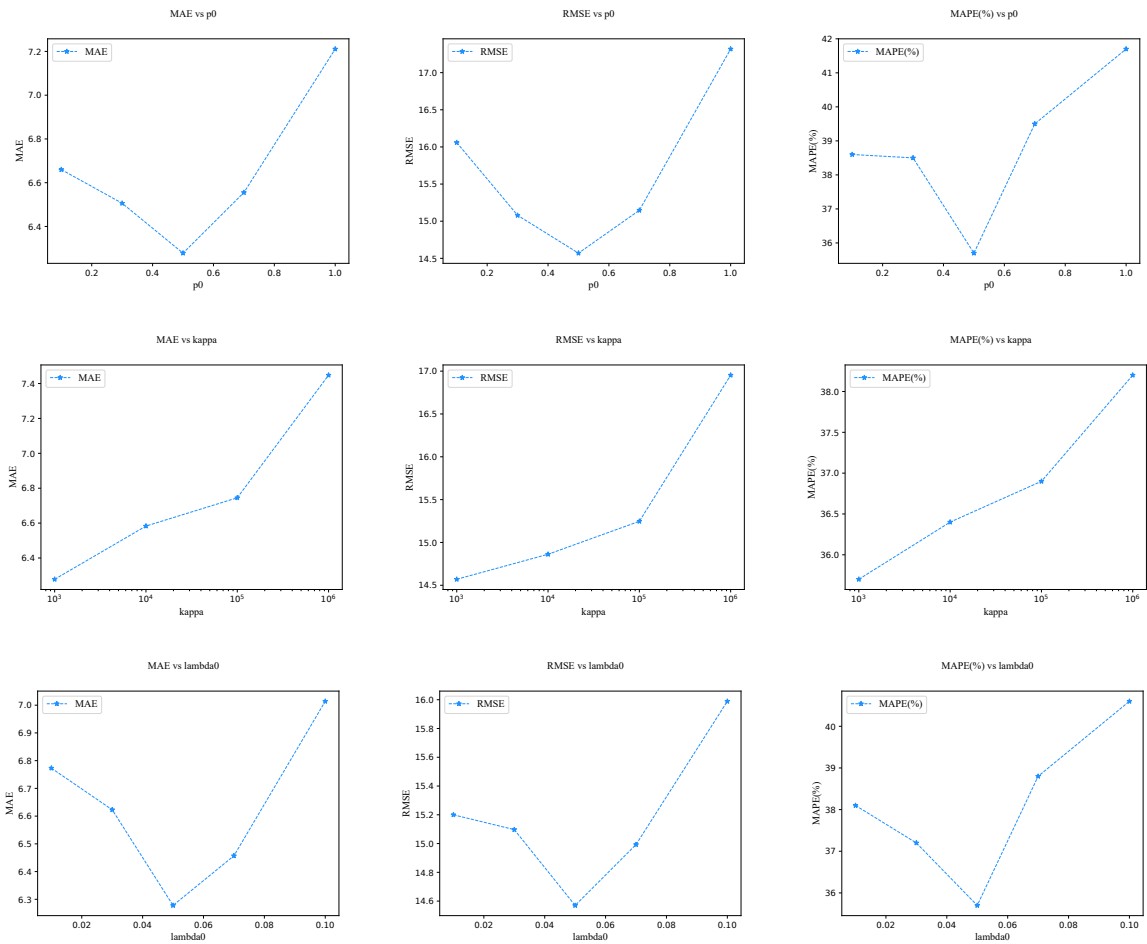

*Figure 4.* Hyperparameter sensitivity on NYC

## B. Hyperparameter sensitivity visualization

To determine the best hyperparameters of our SynEVO and support the completeness of our experiments, we varied base Dropout factor $p_0$ in Eq.10 from $\{0.1, 0.3, 0.5, 0.7, 1\}$, base weight decay coefficient $\lambda_0$ in Eq.11 from $\{0.01, 0.03, 0.05, 0.07, 0.1\}$, and distance threshold $\kappa$ from $\{1 \times 10^3, 1 \times 10^4, 1 \times 10^5, 1 \times 10^6\}$. The experimental results are displayed in Fig.4-Fig.7. 1) Fig.4 shows that on NYC, the performance of SynEVO firstly increases with the increase of $p_0$ and $\lambda_0$ while drops when $p_0 \geq 0.5, \lambda_0 \geq 0.05$. This result demonstrates that on NYC, if $p_0$ and $\lambda_0$ are too small, the model is initially too complex so that it fails to learn from easy to hard, thus trapped in a local optimum. If $p_0$ and $\lambda_0$ are too large, the model will fail to be sufficiently evolved and cannot capture enough commonality of input data. 2) For the distance threshold $\kappa$, if it's extremely small, it means no new domain is allowed into SynEVO, which will violate the principle of 'harmony with diversity for collective intelligence'. If it's too large, then much noise will be introduced to reduce the performance of SynEVO, polluting the commonality of the trained data.

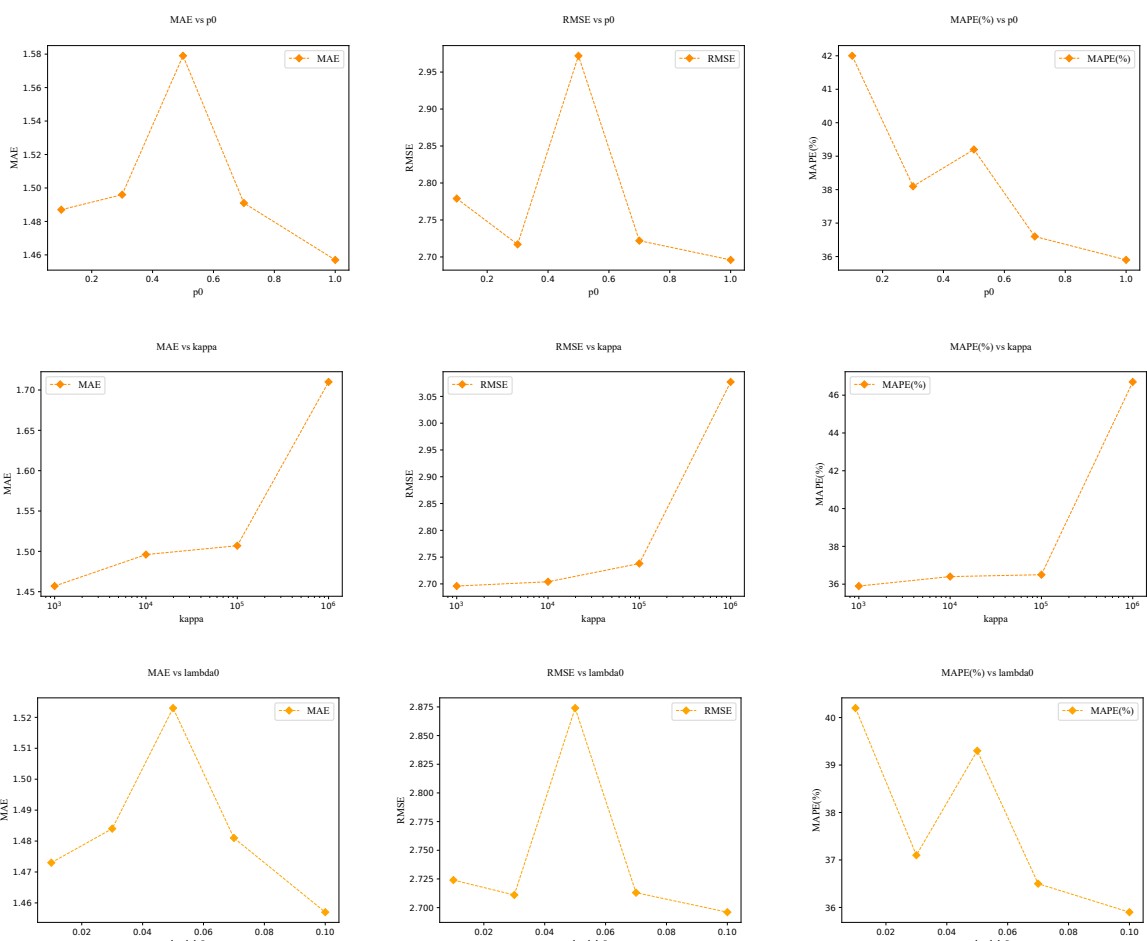

*Figure 5.* Hyperparameter sensitivity on CHI

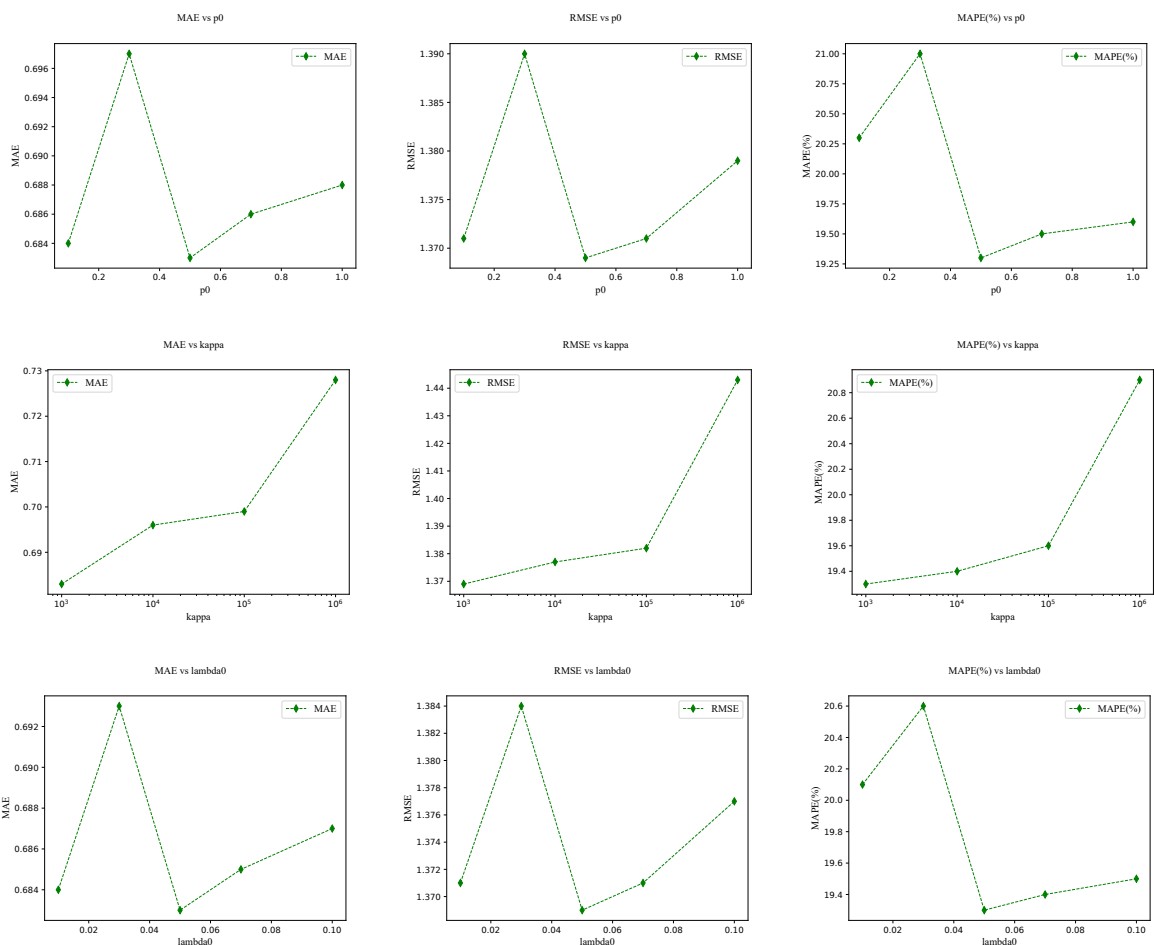

*Figure 6.* Hyperparameter sensitivity on SIP

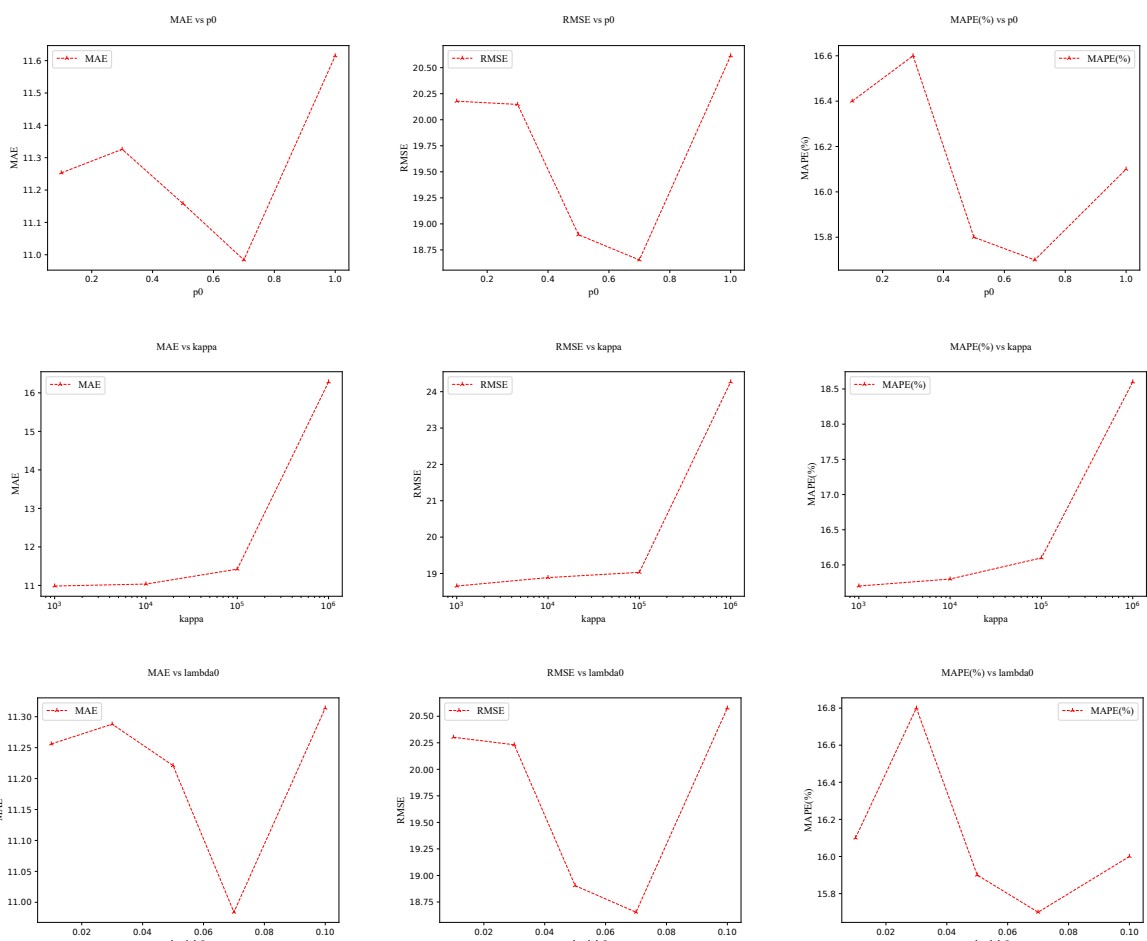

*Figure 7.* Hyperparameter sensitivity on SD

