# OpenReview forum: "SynEVO: A neuro-inspired spatiotemporal evolutional framework for cross-domain adaptation"
_ICML.cc/2025/Conference — ICML 2025 spotlightposter_

### Official Review · Reviewer_ci9Q · 2025-03-03

**Overall Recommendation:** 4

**Summary:**

This work mainly focuses on observation, i.e., deep learning can imitate the neuroscience mechanism to expand the information boundaries through theoretically and empirically cross-domain collective intelligence learning. Then, drawing from neuroscience, this research introduces a synapse-inspired evolutional spatiotemporal network, which facilitates cross-domain knowledge sharing and aggregation. This approach involves three submodules: curriculum learning-based sample group ordering, complementary dual learners, and an adaptive dynamic coupler to capture common intelligence and cross-domain task-dependent patterns. Experiments show that collective intelligence increases the model's generalization capacity under both source and temporal shifts by at 0.5% to 42%, including few-shot and zero-shot transfer.

**Claims And Evidence:**

Yes

**Essential References Not Discussed:**

I have not found any further references to be included.

**Ethical Review Concerns:**

No ethical concerns.

**Experimental Designs Or Analyses:**

Yes. I think the experiment design can be the highlight of this research. The authors design various experiments to verify the effectiveness of such a framework, including designs on cross-source and cross-temporal domain learning for verifying whether enhancing collective intelligence, the comparison against mainstream existing models, and ablation studies for component disentanglements. In addition, three detailed analyses via three typical cases during training are also provided to better understand the framework working flows.

**Methods And Evaluation Criteria:**

Yes. The effectiveness of the proposed solution has been demonstrated by information theories and neuroscience principles with theoretical guarantee. Also, the extensive comparison and ablation studies show improvements against baselines and ablative variants on various datasets.

**Other Comments Or Suggestions:**

1.	In Proposition 3.1, line 133 and Appendix A, line 557, there is a typo: ‘spatiotemporal’ instead of ‘spatiotemoporal’.

2.	In Proposition 3.1, line 134-135, there is a typo: ’there must be shared’ instead of ‘the there must share’

**Other Strengths And Weaknesses:**

Strengths:
1.	A new and pioneering research problem for evolutional learning framework. This research provides a fresh perspective from neuroscience to facilitate the deep model generalization.
2.	Valid techniques and solutions. The proposal focused on imitating the human learning process via various principles and improving the collective intelligence from different domains where similar but different sample groups are sequentially fed into the learning pipeline.
3.	Good structure and experiments. This paper is well-structured with intuitive figure illustrations. The ingenious experiment designs can illustrate the proposal’s effectiveness.

Weakness:
1.	In your experiment, since you use spatiotemporal data, why do you only design temporal adaptation and source adaptation while skipping spatial adaptation?
2.	Several typos to be corrected, please see my ‘Other comments or suggestions’.

**Questions For Authors:**

1.	Since you use spatiotemporal data in your experiment, why do you only design temporal and source adaptation while skipping spatial adaptation?

2.	Several typos to be corrected, please see my ‘Other comments or suggestions’.

**Relation To Broader Scientific Literature:**

This paper contributes to the inter-disciplinary neuroscience, deep learning, and urban science fields. It develops the new techniques of an evolutionary deep learning framework to facilitate gaining collective intelligence among diverse urban data via a set of neuroscience principles, which is associated with the following literature.

a) Xu F, Zhang J, Gao C, et al. Urban generative intelligence (ugi): A foundational platform for agents in embodied city environment[J]. arXiv preprint arXiv:2312.11813, 2023.

b) Feng J, Du Y, Liu T, et al. Citygpt: Empowering urban spatial cognition of large language models[J]. arXiv preprint arXiv:2406.13948, 2024.

c) Bassett D S, Sporns O. Network neuroscience[J]. Nature neuroscience, 2017, 20(3): 353-364.

**Theoretical Claims:**

Yes. In Appendix A, the author proves the correctness of Proposition 3.1 in section 3 based on the theories of information such as information entropy and mutual information, and I think that is roughly reasonable for the claim of Prop 3.1 (cross-domain learning to gain collective intelligence).

---

> ### Author Rebuttal · Authors · 2025-03-31
>
> Dear reviewer ci9Q,
>
> We deeply appreciate the time and efforts you have invested in reviewing our paper. We are honored to receive your recognition of the novelty and theoretical contribution. Your comprehensive feedback is helpful in guiding our revisions.
>
> **W1. Why skip spatial adaptation**
>
> Urban space is relatively static. Over a long period of time, data distribution and data sources will switch and change, but urban space will remain relatively invariant. The change of urban space can be divided into two aspects: 1) the land use of urban space, e.g., the urban land increases. 2) the model transfer among cities. Actually, our research paper focuses on the same urban system, and the scenarios of urban space expansion and cross-city model transfer may be out of the research scope of this work. Instead, we mostly focus on the collective intelligence in the same system, i.e., the data temporal domain distribution shift or the source domain shift, which is more usual in real-world practices. We study the problem of how spatiotemporal learning models adaptively evolve with data by capturing commonality and transfer the regularities to new scenario, so as to achieve the ability of learning whenever new data arrives. For abovementioned urban expansion or cross-city transfer scenarios, we can still take the core idea of this work, reordering new sample groups and make the elastic container grow based on SynEVO to quickly optimize and fine-tune existing models, realizing rapid transfer across different spaces.
>
> **W2&Q1Q2. A list of typos**
>
> Thanks for your careful check of our paper and we have thoroughly corrected the spelling mistakes and typos. The detailed correction are listed as below.
>
> 1) Line 133 and 557, 'spatiotemoporal' -> 'spatiotemporal';
>
> 2) Line 134-135, 'the there must shre' -> 'there must be shared';
>
> 3) Line 380-381, 'signifcant' -> 'significant'.
>
> Thanks again for your expert review of our paper and we will incorporate the discussions and supplement more close literature to facilitate the validness of our paper.
>
> Authors of Paper 5506

---

> > ### Comment · Reviewer_ci9Q · 2025-04-03
> >
> > I appreciate the authors for thoroughly addressing the previous concerns. Considering the author's rebuttals and the positive assessments from other reviewers, I confirm that I lean toward acceptance for this paper.

---

> > > ### Author Response · Authors · 2025-04-06
> > >
> > > Dear Reviewer ci9Q,
> > >
> > > We are extremely delighted to learn that you keep leaning toward acceptance for our work. Your positive evaluation and constructive insights have provided us with invaluable guidance for future research. We are truly grateful for your time and expertise in reviewing our work
> > >
> > > Thank you once more for your generous support and recognition!
> > >
> > > Authors of paper 5506

---

### Official Review · Reviewer_n3PZ · 2025-03-08

**Overall Recommendation:** 4

**Summary:**

This paper, theoretically examines strategies for increasing information boundaries through cross-domain collective intelligence learning and introduces SynEVO, a synaptic evolutionary spatiotemporal network designed to enable cross-domain knowledge sharing and aggregation by addressing model independence constraints. The comprehensive experiments including ablation studies and detailed analysis are provided.

## update after rebuttal
I appreciate the authors' response. I will maintain my positive score.

**Claims And Evidence:**

Yes. The claims are well-supported with both theory principles and empirical experiments.

**Essential References Not Discussed:**

No

**Experimental Designs Or Analyses:**

Yes. This research benefits from their thoughtful experimental designs on three folds. 1) Extensive dataset collection. The authors have collected spatiotemporal data with various domains from four cities, especially a large temporal scale set on SD from LargeST. 2) The cross-domain transfer design for model verification on its claims. 3) The detailed analysis on three aspects supplements more information on how the model operate in a micro and case-by-case perspective.

**Methods And Evaluation Criteria:**

Yes. This proposal makes sense based on neuroscience and information theory of information bottleneck, which contributes to transfer learning and multi-task learning.

**Other Comments Or Suggestions:**

Actually, in Introduction part, how diverse aspects of human learning facilitate the deep neural network for cross-domain learning and transfers requires clarification in a holistic perspective.

**Other Strengths And Weaknesses:**

Strengths:
1. This research paper is novel at new task definition on spatiotemporal cross-domain transfer among both temporal domain and source domain.
2. This paper is well-organized with clear problem, challenges and solutions, along with good figure illustration and detailed experiment designs.
3. The proposed techniques is overall novel and provide new insights into deep neural network designs with mechanisms of human brains.

Weaknesses
1. In line 149, the authors have mentioned ‘in harmony with diversity’, but this concept lacks explanations. I think you are required to elaborate this phrase to moderate the whole sentence and context.
2. In Section 5.6, the authors have mentioned that ‘domains from the same source are not necessarily next to each other in the ordered sequence S’, but this sentence is quite confused and requires efforts with further explanation.

**Questions For Authors:**

1. The authors should clarify more on ‘in harmony with diversity’ and ‘domains from the same source are not necessarily next to each other in the ordered sequence S’.
2. Please elaborate how diverse aspects of human learning facilitate the deep neural network for cross-domain learning and transfers requires clarification in a holistic perspective.

**Relation To Broader Scientific Literature:**

This paper falls onto the application of both neuroscience and improvement of deep neural networks, which is inherited by complementary learning [1], and curriculum learning [2].
[1] McClelland J L, McNaughton B L, Lampinen A K. Integration of new information in memory: new insights from a complementary learning systems perspective[J]. Philosophical Transactions of the Royal Society B, 2020, 375(1799): 20190637.
[2] Matiisen T, Oliver A, Cohen T, et al. Teacher–student curriculum learning[J]. IEEE transactions on neural networks and learning systems, 2019, 31(9): 3732-3740.
But this work inherits such idea and improve it to spatiotemporal learning in several insights.

**Theoretical Claims:**

Yes. In Appendix A, the author applicates the theory of mutual information and information entropy to measure the information in order to prove that the information is increasing during cross-domain adaptation which is claimed in Proposition 3.1. It seems OK for such proposition.

---

> ### Author Rebuttal · Authors · 2025-03-31
>
> Dear reviewer n3PZ,
>
> Thanks for your constructive feedback of our research. Your valuable advice contributes a lot to our work.
>
> **W1&Q1(1). Explanations of 'harmony with diversity'.**
>
> The concept of 'harmony with diversity' in our paper means that in order to build a generic spatiotemporal learning framework, there should be a certain degree of similarity (correlation) between sample groups or tasks, but they are not completely the same. To this end, a commonality learning container can be trained to capture the common patterns and enable distinguished representation learning from personal patterns. Then our learning framework can quickly transfer this commonality patterns to new tasks and obtain rapid generalization. We will supplement more clarification to improve the conciseness of such phrase.
>
> **W2&Q1(2). Explanations of ‘domains from the same source are not necessarily next to each other in the ordered sequence S’ in section 5.6.**
>
> This sentence means that the sample groups from the same source (intra-source sample groups) do not necessarily show greater similarity than sample groups of different sources (inter-source sample groups). This sentence is the interpretation and further discussion of Fig.3(a) in our paper. In the figure, we find sample groups [2,0], [2,1], [2,2] and [2,3] from the second source domain are between the first source domain samples [1,0] and [1,3], which indicates above phenomenon.
>
> **Q2. Elaborate how diverse aspects of human learning facilitate the deep neural network for cross-domain learning and transfers requires clarification in a holistic perspective.**
>
> Neuroscience reveals how brain structure impacts human cognitive and behaviors. The core of this paper is to design the learning process of cross-domain knowledge generalization by imitating neuroscience mechanisms from human brain. Specifically, our paper is divided into two aspects: **(1) curriculum learning** and **(2) synapse structure with elastic neural networks**.
>
> **Curriculum learning** reveals the rule of human brain learning. People usually start to learn from simple tasks, and then learn to master more complex skills with increasing difficulty of learning tasks. Actually, the learning efficiency of such practice is higher than directly resolving new difficult problems [1,2]. To this end, task ordering from easy to difficult and the similarity between tasks are important to facilitate the transfer, which is emphasized in Sec.4.2 of our paper.
>
> **Synapse structure.** From the perspective of human brain evolution, information in the human brain is transmitted through the **synapse structure**. This information passes through the synapse and forms an effective memory. At this time, if a new task needs to be learned, the stored information (i.e., memory in brain) becomes active, the presynaptic neurotransmitter will be released according to the correlation between tasks, so as to enable the original learned knowledge to be effectively transferred to learning process of new task. Thus, the long-term stable memory and new knowledge learning can supplement with each other, i.e., memory is transferred to empower learning while learning (mastering knowledge) expands and enriches the content of memory.
>
> Thanks again for taking time to review our paper and we will incorporate above deeper analysis and explanations of concepts into our paper.
>
> [1] Wang X, Chen Y, Zhu W. A survey on curriculum learning[J]. IEEE transactions on pattern analysis and machine intelligence, 2021, 44(9): 4555-4576.
>
> [2] Blessing D, Celik O, Jia X, et al. Information maximizing curriculum: A curriculum-based approach for learning versatile skills[J]. Advances in Neural Information Processing Systems, 2023, 36: 51536-51561.
>
> Authors of Paper 5506

---

### Official Review · Reviewer_esVG · 2025-03-08

**Overall Recommendation:** 4

**Summary:**

Drawing from neuroscience, this paper presents a theoretical investigation into methodologies for expanding information boundaries via cross-domain collective intelligence learning. The authors propose SynEVO, a synaptic evolutionary spatiotemporal network architecture. The framework employs a sample order reorganization strategy to emulate curriculum learning processes observed in human cognition, coupled with the implementation of two synergistic learning components, where these modules can cooperate with each other to enable model evolution and maintain a clear separation among domain-specific characteristics. The experiments are conducted to verify the effectiveness of the proposal.

**Claims And Evidence:**

Yes. Most of them are clear. But in Lemma 1 (Eq.(8)), are there any references to support such Lemma, and then how to derive the Eq.(10)?

**Essential References Not Discussed:**

There must be some references for citation associated with Lemma 1.

**Experimental Designs Or Analyses:**

Yes.
1) The authors make comparison among baselines like STGCN, STGODE.
2) The author performs ablation studies to uncover the significance of each module (REO, Ela, PE) and display the results in Table 4.
3) The author makes some detailed analyses, including analyses of sample group sequences, observed quick adaptation via loss behavior and effective zero-shot adaptation.

**Methods And Evaluation Criteria:**

Yes. The proposed method seems make sense and resolve the issue of gaining collective intelligence in cross-domain transfer tasks for urban prediction.

**Other Comments Or Suggestions:**

-Please show more explanations or analyses of the relationship between neuroscience and transfer learning in your paper since it’s not clearly clarified in the submission.
-How can you derive Eq (8), is there any supports for it?

**Other Strengths And Weaknesses:**

Overall, this research is somewhat novel with key insights of both neural network designs and how to learn from human knowledge acquiring mechanism. In detail, I found the detailed technical strengths as below,
1.The authors devise a sample group learning order via gradients similarity, diving into the essence of how the deep network updates, thus imitating the learning process from easy to difficult.
2.The authors imitate neuron learning scheme to catch the shared commonality and information among data contents which is essential to cross-domain adaptation.
3.The constructed dual learners (elastic common container and personality extractor) cooperate with each other are relatively novel, where it includes a common container to catch the commonality and a personality extractor to judge the difference, reducing the pollution of inappropriate data. This structure reflects the idea of cooperation and complementary.
Weaknesses
1.LLM is popular nowadays, why not try to applicate it to empower your model?
2.Spatiotemporal data can be divided into spatial data and temporal data. However, I didn’t see the analyses or experiments of the term of spatial. For example, cross-spatial domain adaptation ought to be performed.
3.Some confused equations, e.g., How can you derive Eq (8), is there any supports for it?

**Questions For Authors:**

1.Why gradients can determine the training order? Please show some more reasons.
2.Please show more explanations or analyses of the relationship between neuroscience and transfer learning in your paper since it’s not clearly clarified in the submission.
3.How can you derive Eq (8), is there any supports for it?

**Relation To Broader Scientific Literature:**

This research involves the neuro-related learning into spatiotemporal forecasting, which I think is related to brain-inspired continual learning and complementary learning systems for collecting and gaining collective intelligence. The related researches can be found as below,
1)Van de Ven G M, Siegelmann H T, Tolias A S. Brain-inspired replay for continual learning with artificial neural networks[J]. Nature communications, 2020, 11(1): 4069.
2)Wang L, Zhang X, Su H, et al. A comprehensive survey of continual learning: Theory, method and application[J]. IEEE Transactions on Pattern Analysis and Machine Intelligence, 2024.
3)Wang L, Zhang X, Yang K, et al. Memory replay with data compression for continual learning[J]. arXiv preprint arXiv:2202.06592, 2022.

**Theoretical Claims:**

Yes. The author proves Proposition 3.1 shows that the collective intelligence can be obtained by deriving the larger information entropy via mutual information computation.

---

> ### Author Rebuttal · Authors · 2025-03-31
>
> Dear reviewer esVG,
>
> Thanks for your constructive comments on our research.
>
> **W1. Why not use LLM?**
>
> LLM is popular to empower diverse applications but it is more specific to language processing and generation tasks. The reasons for not using LLM in this research are two-fold.
>
> (1) The core of this research is data-adaptive model evolution, which is essentially a new training paradigm. Our model can be coupled with other deep learning models to achieve efficient data-adaptive model updates. This research is orthogonal to the study of large models.
>
> (2) In fact, studies have demonstrated that smaller models are more suitable for numerical series-level predictions [1], while LLMs tend to excel in language generation tasks including question&answer [2] and planning [3].
>
> In the future, it is interesting to take LLM as an agent to facilitate the learning process and then guide and schedule the adaptation and evolution process. Thanks again for your insightful question.
>
> **W2. Why skip cross-spatial domain adaptation?**
>
> Thanks for your valuable question. We list the reasons in the **W1&Q1 response to Reviewer ci9Q**
>
> **W3&Q3 How to derive Eq.(8) and from Eq.(8) to Eq.(10)**
>
> Our analysis is based Ref.[4] and [5].
> Ref.[4] proposes an exponential model of the transmitter release probability:
>         $P_r=1-e^{[Ca^{2+}]^n/K}$
> where K combines parameters such as calcium binding rate and vesicle fusion efficiency, which affect information absorption in synapses. This formula explains the randomness of the combination of calcium ions with synaptotagmin.
>
> After that, Ref.[5] directly measured the relationship between calcium concentration and transmitter release rate, and verified the applicability of the above exponential model.  Finally, we reduce ${{[{Ca}^{2+}]}^n/K}$ to a variable $\tau$ to obtain Eq(8).
>
> In our task reordering module, tasks are ordered from easy to difficult, by ranking the gradient from small to large. Since the smaller the dropout factor p, the more active the model is and the easier to achieve difficult tasks, our dropout factor p needs to decrease as the gradient gets larger. Hence, we can achieve Eq.(10). We let the exponent be $l(d_c)-d_{max}$ to make $0<p_c(d_c)\le1$.
>
> **Q1.Why gradients determine the order?**
>
> For each parameter w in the neural network, each step of parameter update is related to its own gradient, i.e.
> $W_k=W_{k-1}-\eta\nabla_k W=W_{k-1}-\eta∂E/∂W$
> where $W_k$ is the parameter of k-th step, $\eta$ is the learning rate, $\eta∂E/∂W$ is the gradient of training loss to W.
>
> From the equation above, we can see that the closeness of the model to learning samples (data) can be described by the gradient of each iteration. Larger gradients indicate the current model is farther from the real function which perfectly fits the data, then the learning process is harder. To this end, gradients can determine the training order of sample groups.
>
> **Q2. Relationship between neuroscience and transfer learning**
>
> The goal of transfer learning is to effectively transfer the knowledge of model A to a relevant task B. During the process, similarity counts. Based on theories of curriculum learning, humans master skills better when learning from easy to hard. During the learning process, similarity also exists between tasks. From the perspective of human brain evolution, information is transmitted through synapses in the human brain. This information passes through the synapse and forms a long-time stable memory. When facing similar new tasks, the brain releases presynaptic neurotransmitters based on the similarity between tasks, transferring the original knowledge to the new task. During the process, memory and learning can complement with each other, i.e., new well-learned knowledge can be expanded into memory, while existing memory can be retrieved when new knowledge is being learned.
>
> We are truly grateful for your meticulous review and constructive comments on our manuscript. They have significantly contributed to enhancing the clarity and rigor of our work. We will add above deeper discussions into our next version.
>
> [1] Tan M, et al. Are language models actually useful for time series forecasting?[J]. Advances in Neural Information Processing Systems, 2024, 37: 60162-60191.
>
> [2] Feng P,  et al. AGILE: A Novel Reinforcement Learning Framework of LLM Agents[C]//The Thirty-eighth Annual Conference on Neural Information Processing Systems.
>
> [3] Ni H, et al. Planning, Living and Judging: A Multi-agent LLM-based Framework for Cyclical Urban Planning[J]. arXiv preprint arXiv:2412.20505, 2024.
>
> [4] Bertram, Richard, Arthur Sherman, and ELIS F. Stanley. "Single-domain/bound calcium hypothesis of transmitter release and facilitation." Journal of Neurophysiology 75.5 (1996): 1919-1931.
>
> [5] Schneggenburger, Ralf, and Erwin Neher. "Intracellular calcium dependence of transmitter release rates at a fast central synapse." Nature 406.6798 (2000): 889-893.
>
> Authors of Paper 5506

---

> > ### Comment · Reviewer_esVG · 2025-04-04
> >
> > I appreciate the authors for addressing my concerns in detail, and I decided to raise my score.

---

> > > ### Author Response · Authors · 2025-04-06
> > >
> > > Dear Reviewer esVG,
> > >
> > > We are truly grateful for your thoughtful evaluation and for increasing the score for our work. Your valuable feedback and recognition mean a great deal to us, and we are truly honored by your appreciation. We will continue to strive for excellence in our research and contributions to the field.
> > >
> > > Thanks again for your time and effort in reviewing our work!
> > >
> > > Authors of paper 5506

---

### Official Review · Reviewer_Nb7e · 2025-03-13

**Overall Recommendation:** 3

**Summary:**

This paper introduces SynEVO, an interesting neuro-inspired spatiotemporal evolutional framework designed for cross-domain adaptation in spatiotemporal learning. The core idea is to enhance knowledge transfer and model evolution by mimicking synaptic plasticity and neurotransmitter mechanisms from neuroscience.

**Claims And Evidence:**

Most claims are well-supported by theoretical and empirical evidence:
1. Cross-domain learning increases information capacity – Supported by information-theoretic proof.
2. SynEVO improves generalization – supported by experiments on four datasets, though results may be dataset-dependent.
3. Superior efficiency (21.75% memory cost) – GPU usage comparisons validate this.
4. Neuro-inspired synaptic evolution – Conceptually compelling and very intersting.

**Essential References Not Discussed:**

NA

**Experimental Designs Or Analyses:**

The use of four real-world datasets (NYC, CHI, SIP, SD) provides a diverse and realistic evaluation.
Metrics such as MAE, RMSE, and MAPE are standard for assessing spatiotemporal prediction accuracy.
Ablation studies effectively isolate contributions of key components.
However, given the general nature of the approach, incorporating additional tasks could further strengthen the study.
Moreover, related areas such as transfer learning, optimizer design, and even evolutionary algorithms offer valuable directions for discussion. A more thorough theoretical and empirical analysis could enhance the justification for the paper's design choices. Nevertheless, the overall contribution is good.

**Methods And Evaluation Criteria:**

1. Evaluation is robust, utilizing four real-world datasets (NYC, CHI, SIP, SD) and standard metrics (MAE, RMSE, MAPE) with comparisons against strong baselines.
2. Ablation studies validate key components, but broader application and more analysis of training efficiency would strengthen the evaluation.

**Other Comments Or Suggestions:**

see above

**Other Strengths And Weaknesses:**

see above

**Questions For Authors:**

NA

**Relation To Broader Scientific Literature:**

This paper is inspired by neuroscience, particularly ideas related to how the brain gradually learns and adapts by evolving its connections over time. Similar principles have been explored in studies on how humans accumulate knowledge and transfer learning across different tasks. The proposed approach also relates to continual learning in LLM/VLM, especially with the rise of large models. As models are increasingly required to adapt to new tasks without forgetting previous knowledge, frameworks like SynEVO offer a potential way to improve long-term adaptation and efficient knowledge retention.

**Theoretical Claims:**

The author presents an information-theoretic perspective, which is both interesting and novel.

---

> ### Author Rebuttal · Authors · 2025-03-31
>
> Dear reviewer Nb7e,
>
> Thanks for your encouraging comments!
>
> **Relations between SynEVO and transfer learning, optimizer design, evolutionary algorithms**
>
> **Transfer learning** is also a freeze-finetune mechanism where finetune is varied and specific to the problem itself. It does not involve active evolutions under different data distributions, while for SynEVO, it actively and continuously evolves by updating parameters when new data comes.
>
> **Evolutionary algorithm** tends to evolve by adaptive fusion among peer models. It can be viewed as a passive evolution, while SynEVO updates spontaneously according to data.
>
> **Optimizer design** includes various gradient-based adaptiveness and regularizations. In our work, we take gradients to measure correlations between model and data. We let it reorder sample groups to imitate curriculum learning of brain. Besides, we enable the regularization coefficient $\lambda$ to change dynamically with increasing learning capacity. These schemes are devised for easy optimization in our evolution contexts.
>
> **Detailed empirical analysis and training efficiency**
>
> **(1) More empirical analysis on task reordering.**
>
> We supplement the experiments of reordering tasks from **hard to easy** to emphasize the importance of training order. The results of errors are listed in the order of MAE/RMSE/MAPE.
>
> NYC: 7.217/18.807/0.415, CHI: 1.632/3.066/0.405;
>
> SIP: 0.705/1.394/0.208, SD: 11.636/19.789/0.163.
>
> The reversed order falls into inferior performances, which emphasizes the significance of reordering the tasks from easy to difficult in our design. Moreover, above hard-to-easy performances are better than random ordering of SynEVO-REO, which may be attributed to common relations between neighboring tasks as they are ordered even the reverse one.
>
> **(2) Design of commonality extraction.**
>
> a)	If iterative learning for commonality removed, only train and test on one dataset. Results are listed in order of MAE/RMSE/MAPE:
>
> NYC: 8.201/19.090/0.423, CHI: 1.554/2.887/0.385;
>
> SIP: 0.711/1.412/0.216, SD: 13.128/20.890/0.220.
>
> b) We set a different neural expansion rate $p_c$, e.g., $p_c=p_0/l(d_c)$
>
> NYC: 7.213/16.310/0.422, CHI:1.550/2.765/0.367; SIP: 0.705/1.399/0.207, SD:11.944/19.599/0.168.
>
> With above results, we can conclude our commonality learner and the setting of p are reasonable and empirically justified.
>
> **(3) Training efficiency.**
>
> Our training process can be 4 parts. I. Model warm-up with existing data. II. Complementary dual learners with elastic common container.  III. Contrastive learning to obtain distinguished patterns. IV. Couple elastic common container and personality extractor to train on new data.
>
> **Comparison.** On NYC, the above 4 parts cost about 800s, 795s, 545s and 111s respectively and the total time cost is about 2251s. The total time costs of baselines: AGCRN:1896s, ASTGCN:1884s, GWN:2233s, STGCN:823s, STGODE:3515s, STTN:1910s, CMuST: 2817s. Given increased generalization capacity (MAE reduced 42% at most) and inference efficiency (GPU cost reduced 78% at most), thus training costs are tolerable. Specifically, in our ablation studies, we can further confirm the importance of each module as shown in Tab.4.
>
> **Further theoretical analysis**
>
> **(1) Information theory.** Based on Appendix A, collective intelligence within a system increases with the input of data, i.e.,
> $H(X_i|X_1,X_2,\ldots,X_{i-1})> H(X_{i+1}|X_1,X_2,\ldots,X_i)$
> Thus, we can take full advantage of the collective intelligence in the system to empower task generalization.
>
> **(2) Synaptic structure** is the medium of information transmission in human brain. The probability of synaptic neurotransmitter release can be defined as, $P_r=P_0(1-e^{-\tau})$, $\tau$ is the successive activeness difference between apre-synaptic-neuron and after-synaptic-neuron.
>
> Since the probability is exponentially correlated with the parameter $\tau$, we envisage placing the gradient in an exponential position to map it. In task reordering, the tasks are ordered by gradients from small to large. Considering that decreasing dropout factor $p$ increases model activeness and enhancing its ability to cope with complex tasks, we further deduce Eq.(10). Meanwhile, we let the exponent be $l(d_c)-d_{max}$ to ensure $0<p_c(d_c)\le1$.
>
> **Broader application.**
>
> Our model can be generally nested within other neural networks. For example, in geological prospecting, the available shallow-layer data and precious deep-layer information share both commonality and personalization. It is applicable to utilize our evolvable "data-model" collaboration to decouple the invariant and variable patterns, and reconstruct the OOD distribution with new patterns. More broadly, our solution can also be extended to dynamic systems such as molecular interactions, agent collaborations, with necessary adaptation.
>
> Thanks again for your advice and we will include additional results and discussions into our next version.
>
> Authors of Paper 5506

---

### Decision · Program_Chairs · 2025-05-01

**Decision:**

Accept (spotlight poster)

**Comment:**

This paper proposes SynEVO, a neuro-inspired framework for spatiotemporal cross-domain adaptation. Drawing from neuroscience concepts like curriculum learning and synaptic evolution, the framework enables dynamic model growth and knowledge transfer across domains. It combines task reordering, dual learners for disentangling common and domain-specific representations, and an adaptive coupler for continual evolution.

The paper’s strengths lie in its novelty, theoretical grounding, and comprehensive empirical evaluation. It demonstrates consistent improvements across multiple datasets and scenarios, with up to 42% gains in generalization. Ablation studies and runtime comparisons support the design, and the authors provide thoughtful rebuttals that clarify theoretical points and address reviewer concerns effectively.

The main weaknesses are minor. Some mathematical derivations could use more justification, and the paper does not evaluate spatial adaptation explicitly. However, the authors justify this focus and explain the broader applicability of the method.

Overall, this is a creative and well-executed contribution that connects neuroscience principles with machine learning and offers a useful direction for adaptive spatiotemporal modeling. I recommend acceptance.